# Mass Spectrometry Advancements and Applications for Biomarker Discovery, Diagnostic Innovations, and Personalized Medicine

**DOI:** 10.3390/ijms25189880

**Published:** 2024-09-12

**Authors:** Ahrum Son, Woojin Kim, Jongham Park, Yongho Park, Wonseok Lee, Sangwoon Lee, Hyunsoo Kim

**Affiliations:** 1Department of Molecular Medicine, Scripps Research, San Diego, CA 92037, USA; ahson@scripps.edu; 2Department of Bio-AI Convergence, Chungnam National University, Daejeon 34134, Republic of Korea; woojin1544@gmail.com (W.K.); 975pjh@gmail.com (J.P.); kmalrpkr13@gmail.com (Y.P.); wonseogi46@gmail.com (W.L.); sanguni088@gmail.com (S.L.); 3Department of Convergent Bioscience and Informatics, Chungnam National University, 99 Daehak-ro, Yuseong-gu, Daejeon 34134, Republic of Korea; 4Protein AI Design Institute, Chungnam National University, 99 Daehak-ro, Yuseong-gu, Daejeon 34134, Republic of Korea; 5SCICS, 99 Daehak-ro, Yuseong-gu, Daejeon 34134, Republic of Korea

**Keywords:** liquid chromatography–mass spectrometry, clinical chemistry, biomarker discovery, personalized medicine

## Abstract

Mass spectrometry (MS) has revolutionized clinical chemistry, offering unparalleled capabilities for biomolecule analysis. This review explores the growing significance of mass spectrometry (MS), particularly when coupled with liquid chromatography (LC), in identifying disease biomarkers and quantifying biomolecules for diagnostic and prognostic purposes. The unique advantages of MS in accurately identifying and quantifying diverse molecules have positioned it as a cornerstone in personalized-medicine advancement. MS-based technologies have transformed precision medicine, enabling a comprehensive understanding of disease mechanisms and patient-specific treatment responses. LC-MS has shown exceptional utility in analyzing complex biological matrices, while high-resolution MS has expanded analytical capabilities, allowing the detection of low-abundance molecules and the elucidation of complex biological pathways. The integration of MS with other techniques, such as ion mobility spectrometry, has opened new avenues for biomarker discovery and validation. As we progress toward precision medicine, MS-based technologies will be crucial in addressing the challenges of individualized patient care, driving innovations in disease diagnosis, prognosis, and treatment strategies.

## 1. Introduction

Mass spectrometry (MS) has emerged as a powerful analytical tool in clinical chemistry, offering unprecedented capabilities for qualitative and quantitative analyses of biomolecules [1,2]. This review examines the growing importance of MS, particularly when coupled with liquid chromatography (LC-MS), in identifying disease biomarkers and quantifying biomolecules and drugs for diagnostic and prognostic purposes [1,3]. The unique advantages of MS technology in accurately identifying and quantifying diverse endogenous and exogenous biomolecules have positioned it as a cornerstone in the advancement of personalized medicine [1,4]. As the field of clinical chemistry evolves, there is an increasing demand for technologies that can serve as reference methods and meet analytical needs with high accuracy and precision. Mass spectrometry, with its high specificity and sensitivity, is poised to fulfill this role, offering solutions to many of the limitations faced by traditional analytical techniques [1,2].

The integration of MS-based technologies into clinical practice has revolutionized the field of precision medicine, enabling a more-comprehensive understanding of disease mechanisms and patient-specific responses to treatments [4,5]. LC-MS, in particular, has demonstrated exceptional utility in the analysis of complex biological matrices, allowing for the simultaneous detection and quantification of multiple analytes with high sensitivity and selectivity [3,6]. One of the key advantages of MS in clinical applications is its ability to provide detailed molecular information, which is crucial for the identification of novel biomarkers and the development of targeted therapies [5]. This capability has been particularly valuable in oncology, where MS-based proteomics and metabolomics have facilitated the discovery of cancer-specific markers and potential therapeutic targets [2,5].

Furthermore, the application of MS in therapeutic drug monitoring has significantly improved patient care by enabling precise dosage adjustments based on individual pharmacokinetics and pharmacodynamics [4]. This personalized approach to drug administration has led to enhanced treatment efficacy and reduced adverse effects, particularly in cases involving drugs with narrow therapeutic windows. The advent of high-resolution mass spectrometry (HRMS) has further expanded analytical capabilities in clinical settings, allowing for the detection of low-abundance molecules and the elucidation of complex biological pathways. This technology has been instrumental in advancing our understanding of disease pathogenesis and in identifying novel therapeutic interventions.

In recent years, the integration of MS with other analytical techniques, such as ion mobility spectrometry, has opened new avenues for biomarker discovery and validation [7]. These hyphenated techniques offer enhanced separation capabilities and structural information, further improving the specificity and sensitivity of clinical analyses [8]. The role of MS in personalized medicine extends beyond biomarker discovery and drug monitoring [4,5]. It has also found applications in the field of pharmacogenomics, where it aids in the identification of genetic variants that influence drug metabolism and responses. This integration of genomic and proteomic data has paved the way for more targeted and effective therapeutic strategies. As we move toward an era of precision medicine, the continued development and refinement of MS-based technologies will be crucial in addressing the complex challenges of individualized patient care. The ability of MS to provide comprehensive molecular profiles of biological samples positions it as an indispensable tool in the ongoing efforts to tailor medical treatments to individual patients.

## 2. Advantages of Mass Spectrometry in Clinical Applications

### 2.1. High Specificity and Sensitivity

Mass spectrometry (MS) provides unparalleled specificity in molecular identification, allowing for the accurate detection and quantification of analytes, even in complex biological matrices [5]. Recent advancements in high-resolution accurate-mass (HRAM) spectrometers, such as time-of-flight MS (TOF MS) and Orbitrap, have significantly enhanced the sensitivity and resolution of MS, facilitating its transition from analytical chemistry laboratories to clinical settings [9]. Techniques like ion mobility MS, which separates ionized molecules based on their mobility in a carrier gas, further improve the resolving power and sensitivity of MS, making it ideal for high-throughput proteomics [10]. Additionally, liquid chromatography–mass spectrometry (LC-MS) has become a preferred analytical technique because of its high sensitivity and broad applicability, with strategies to optimize ionization efficiency and reduce contaminants to enhance its signal-to-noise ratio. Prioritized MS approaches, such as pSCoPE, have increased the depth and dynamic range of protein quantification in single-cell proteomics, demonstrating the ability to quantify low-abundance peptides that are often missed by traditional methods [11]. These advancements underscore the critical role of MS in advancing both clinical diagnostics and research by providing highly specific and sensitive analytical capabilities.

### 2.2. Multiplexing Capabilities

Mass spectrometry (MS) techniques, especially when coupled with liquid chromatography (LC-MS), have emerged as powerful tools for simultaneously analyzing multiple analytes in a single run, enabling comprehensive metabolomic and proteomic profiling [12]. The ability of MS to detect and quantify many thousands of metabolite features simultaneously has revolutionized the field of metabolomics, allowing for in-depth characterization of complex biological samples [3]. In proteomics, MS-based approaches can routinely detect and quantify thousands of proteins, with recent advancements in instrumentation and methodology significantly enhancing the sensitivity and resolution [13]. Multiplexing strategies, such as isobaric tagging, have further improved the throughput and quantitative capabilities of MS-based proteomics, enabling the comparison of protein expressions across multiple samples in a single experiment [12]. These multiplexed approaches not only increase the sample throughput but also improve the measurement precision and reduce missing values in large-scale proteomic studies [13]. In metabolomics, the combination of high-resolution accurate-mass (HRAM) spectrometry with advanced separation techniques, like ion mobility, has greatly enhanced the ability to resolve and identify metabolites in complex mixtures. The integration of targeted and untargeted approaches in both metabolomics and proteomics has expanded the depth and breadth of biological insights that can be gained from MS-based analyses [14]. Despite these advancements, challenges remain in fully characterizing the metabolome and proteome, particularly for low-abundance species and in distinguishing isomers [15]. Ongoing advancements in mass spectrometry (MS) technology, including improvements in ionization efficiency, mass accuracy, and multiplexing capacity, continue to expand the possibilities for large-scale molecular profiling of biological systems. [5].

### 2.3. Versatility

Mass spectrometry (MS) can be applied to a wide range of biomolecules, including proteins, peptides, metabolites, and drugs, making it a versatile tool for various clinical applications [1]. The ability of MS to analyze complex biological samples with high specificity and sensitivity has made it indispensable in clinical diagnostics and research [16]. Recent advancements in MS technology, such as high-resolution accurate-mass (HRAM) spectrometers and tandem MS, have significantly enhanced the detection and quantification capabilities of this technique, allowing for the identification of disease biomarkers and therapeutic drug monitoring [17]. MS-based proteomics, for instance, enables the comprehensive analysis of protein expressions and post-translational modifications, which are crucial for understanding disease mechanisms and developing targeted therapies [3]. Additionally, MS has been instrumental in metabolomics, providing detailed metabolic profiles that can reveal insights into metabolic disorders and potential therapeutic targets. The integration of liquid chromatography (LC) with MS further enhances its versatility by improving the separation and analysis of complex mixtures, thereby increasing the accuracy and reliability of the results [18]. Techniques such as ion mobility spectrometry (IMS) coupled with MS have also been developed to separate ions based on their shape and size, adding another dimension to the analysis and improving the resolution of complex samples [5]. Ion mobility spectrometry (IMS) combined with MS introduces an additional level of separation that relies on the molecular characteristics of shape and size. IMS-MS has demonstrated its worth in the analysis of intricate mixtures in proteomic, metabolomic, and lipidomic applications [19]. The additional separation provided by IMS increases the maximum capacity and enhances the study of isomers and isobars. Despite these advancements, challenges remain in fully characterizing low-abundance species and distinguishing isomers. However, ongoing developments in MS technology continue to push the boundaries of molecular profiling [20]. The continuous evolution of MS, including improvements in ionization efficiency, mass accuracy, and data analysis, ensures its pivotal role in advancing clinical diagnostics and personalized medicine [1].

### 2.4. Isotope Dilution Internal Standardization

Isotope dilution mass spectrometry (IDMS) is a unique feature of MS that allows for highly accurate quantification by compensating for matrix effects, a significant advantage over traditional immunoassays [1]. Matrix effects, which can compromise the sensitivity and selectivity of MS, are mitigated in IDMS through the use of isotopically labeled internal standards that closely mimic the behavior of the analyte during the analytical process [21]. This approach significantly enhances the precision and accuracy of quantitative measurements in complex biological matrices, such as plasma, serum, and urine [1]. The use of IDMS in clinical applications has been shown to provide superior specificity and expanded linear ranges compared to those of traditional immunoassays, making it an indispensable tool for therapeutic drug monitoring and biomarker quantification [22]. Recent advancements in MS technology, including high-resolution accurate-mass (HRAM) spectrometers and tandem MS, have further improved the capabilities of IDMS, enabling more reliable and reproducible results [17]. Calibration practices in clinical MS laboratories have also evolved, with the use of matrix-matched calibrators and stable-isotope-labeled internal standards to mitigate the impacts of matrix effects, ensuring the accuracy and precision of the regression models used for quantification [23]. Despite the technical challenges associated with obtaining isotope-labeled internal standards, the benefits of IDMS in terms of reliability and quality control in targeted proteomic analysis are well documented [24]. The ongoing evolution and optimization of isotope dilution mass spectrometry (IDMS) methods further extend the potential for molecular profiling and clinical diagnostics, underscoring its key role in advancing personalized medicine [3] (Figure 1).

## 3. Applications in Biomarker Discovery and Personalized Medicine

### 3.1. Proteomics and Biomarker Discovery

Mass spectrometry (MS) has become a potent analytical instrument in the fields of clinical chemistry and biomarker discovery, allowing for the comprehensive analysis of intricate biological samples with high sensitivity and specificity. Biomarker research for a variety of diseases, such as cancer, cardiovascular disorders, and neurodegenerative conditions, has been significantly transformed by recent advancements in MS-based proteomics [25]. In this discipline, a variety of critical MS techniques are implemented, such as shotgun proteomics for the comprehensive identification of proteins, targeted proteomics for the precise quantification of specific proteins, and data-independent acquisition (DIA) for the systematic analysis of all the detectable peptides [26]. Understanding disease mechanisms and identifying potential diagnostic and prognostic biomarkers have been significantly enhanced as a result of these methods. MS-based proteomics has been instrumental in the discovery of protein signatures that are associated with tumor progression and treatment responses in cancer research [2]. For instance, recent research has employed DIA to identify plasma protein panels for the early detection of pancreatic cancer [1]. Large-scale plasma proteomics has revealed novel biomarkers for the prediction of incident heart failure in cardiovascular medicine [27]. MS has facilitated the discovery of cerebrospinal fluid biomarkers that reflect pathological brain changes in neurodegenerative diseases, such as Alzheimer’s. Recent research has focused on the validation of blood-based biomarkers as less-invasive alternatives [28].

MS-based metabolomics has uncovered metabolic signatures for the early detection of malignancy and the prediction of cardiovascular risk, in addition to complementing proteomics. Metabolic signatures for early cancer detection have been identified in recent metabolomic investigations, including a urine metabolite panel for bladder cancer screening [29]. MS capabilities for clinical applications are being further enhanced by ongoing technological advancements, including ion mobility spectrometry and high-resolution mass analyzers [30]. Standardizing protocols and validating candidate biomarkers in large, diverse patient cohorts before clinical implementation remain challenging, despite these advancements. The translation of these discoveries into routine clinical practice will be contingent upon the integration of multi-omic data and the development of simplified, automated MS workflows [31]. The potential for incorporating metabolomic biomarkers into established risk prediction models, such as the SCORE2 model for cardiovascular risk, has been recently demonstrated, resulting in a significant improvement in predictive performance. Furthermore, the interpretation of intricate multi-omic datasets is being improved by advancements in bioinformatics and machine-learning techniques, which is enabling the identification of new biomarker panels with enhanced diagnostic and prognostic efficacies [32]. The integration of MS technologies into clinical workflows holds the potential to revolutionize personalized medicine by enabling more precise diagnoses, prognoses, and treatment monitoring across a wide range of diseases as these technologies continue to evolve (Figure 2A).

### 3.2. Therapeutic Drug Monitoring

The gold standard for therapeutic drug monitoring (TDM) has been established by liquid chromatography–tandem mass spectrometry (LC-MS/MS) as a result of its exceptional specificity, sensitivity, and capacity to simultaneously quantify multiple analytes [26,33]. This sophisticated method allows for the precise measurement of drug concentrations, which in turn facilitates the development of personalized dosage adjustments and treatment strategies in a variety of therapeutic areas, such as antiepileptics, immunosuppressants, and cytotoxic agents [9,34]. The efficiency and accuracy of LC-MS/MS methodologies have been further improved by recent advancements in sample preparation techniques, chromatographic separations, and mass spectrometric detection [1,34]. The TDM of kinase inhibitors has garnered significant attention in the field of oncology, as a number of compounds have established exposure–response and exposure–toxicity relationships. This has the potential to enhance the efficacy of treatments and reduce the occurrence of adverse effects [5,35]. The integration of TDM data with pharmacogenomic information and clinical parameters has enabled the development of more comprehensive and personalized treatment approaches, thereby improving patient care and reducing healthcare costs.

MS-based proteomics and metabolomics have emerged as potent instruments for the identification of novel disease markers and the elucidation of underlying biological mechanisms in the field of biomarker discovery [36]. The detection and quantification of thousands of proteins and metabolites in a single analysis have been made possible by high-resolution mass spectrometry techniques, including Orbitrap and time-of-flight (TOF) analyzers, which have provided an unprecedented level of molecular profiling [37]. Data-independent acquisition (DIA) methods have further improved the reproducibility and comprehensiveness of proteomic analyses, thereby enabling the conduct of large-scale biomarker investigations [38]. The sensitivity and specificity of candidate biomarker validation have been enhanced by recent advancements in targeted proteomics, such as parallel reaction monitoring (PRM) and multiple reaction monitoring (MRM) [2]. MS-based methodologies have resulted in the identification of promising biomarkers for the prediction of treatment responses and prognoses and early detection in cancer research [39]. For example, a recent study utilizing DIA proteomics identified a panel of plasma proteins for the early detection of pancreatic cancer [40]. Additionally, MS has facilitated the discovery of cerebrospinal fluid and blood-based biomarkers that reflect pathological brain changes in neurodegenerative diseases. These biomarkers may be used in diagnosing Alzheimer’s and Parkinson’s diseases [5]. The incorporation of MS technologies into clinical workflows has the potential to revolutionize personalized medicine by enabling more precise diagnoses, prognoses, and treatment monitoring across a broad spectrum of diseases as these technologies continue to evolve [26] (Figure 2B).

### 3.3. Endocrinology

The field of clinical endocrinology has been substantially improved by mass spectrometry (MS), which has revolutionized diagnostic and therapeutic strategies by improving the accuracy and specificity of hormone measurements. Gold standards for hormone analysis, including steroid hormone measurements, thyroid function tests, and vitamin D metabolite quantification, have been established by liquid chromatography–tandem mass spectrometry (LC-MS/MS) due to its superior selectivity and multiplexing capabilities compared to traditional immunoassays [41,42]. This technology enables the precise quantification of low-abundance hormones, including dihydrotestosterone, estradiol, and aldosterone, which are frequently difficult to measure accurately with immunoassays [41,43,44]. This has resulted in enhanced diagnostic accuracy and patient outcomes, as the high specificity of MS has revealed inaccuracies in numerous automated immunoassays, particularly under complex physiological conditions [42,44,45]. The reliability of clinical assays has been improved by the standardization and harmonization of hormone measurements, which have been facilitated by recent advancements in MS technology [42]. The advancement of precision medicine in endocrinology has been significantly driven by the integration of MS-based hormone assays into routine clinical practice, enabling more accurate disease diagnoses and management strategies [46,47].

MS-based proteomics and metabolomics have emerged as potent instruments for the identification of novel disease markers and the elucidation of underlying biological mechanisms in the field of biomarker discovery [28]. The detection and quantification of thousands of proteins and metabolites in a single analysis have been made possible by high-resolution MS techniques, including Orbitrap and time-of-flight (TOF) analyzers, which have provided an unprecedented level of molecular profiling [37]. The reproducibility and comprehensiveness of proteomic analyses have been further improved by data-independent acquisition (DIA) methods, which have facilitated the development of large-scale biomarker studies [36]. Parallel reaction monitoring (PRM) and multiple reaction monitoring (MRM), which are recent developments in targeted proteomics, have enhanced the specificity and sensitivity of candidate biomarker validation [48]. The discovery of promising biomarkers for early detection, prognosis, and treatment response prediction has been facilitated by MS-based approaches in cancer research [49]. For example, a recent study that employed DIA proteomics has identified a panel of plasma proteins for the early detection of pancreatic cancer [50]. MS has facilitated the discovery of cerebrospinal-fluid- and blood-based biomarkers that reflect pathological brain changes in neurodegenerative diseases. These biomarkers have the potential to be used in the diagnoses of Alzheimer’s and Parkinson’s diseases [5]. The incorporation of MS technologies into clinical workflows has the potential to revolutionize personalized medicine by facilitating more precise diagnoses, prognoses, and treatment monitoring across a broad spectrum of diseases as they continue to evolve [51] (Figure 2C).

### 3.4. Microbiology

Mass spectrometry (MS) imaging techniques offer the ability to analyze samples in a structured arrangement, providing detailed molecular information with spatial resolution. Matrix-assisted laser desorption/ionization (MALDI) imaging mass spectrometry (MS) is frequently employed to analyze and visualize the spatial distributions of proteins, lipids, metabolites, and pharmaceuticals in tissue slices [52]. The advancements in MALDI imaging technology have made it possible to image individual cells at a spatial resolution of less than 1 μm [53]. DESI imaging enables the direct examination of tissues in their natural state, eliminating the need for a matrix. This technology has considerably facilitated the real-time analysis of tissues during cancer surgery among other applications [54]. Laser ablation electrospray ionization (LAESI) is an alternative technique that enables three-dimensional molecular imaging of plant and animal tissues [55].

Matrix-assisted laser desorption/ionization time-of-flight mass spectrometry (MALDI-TOF MS) has revolutionized clinical microbiology by facilitating the rapid and accurate identification of microorganisms, thereby significantly reducing the time required for pathogen identification and enabling the formulation of more targeted, expedited treatment decisions [56,57]. This method is capable for identifying a diverse array of microorganisms, such as bacteria, fungi, and viruses, by comparing their distinctive protein spectra to those in extensive databases [58]. MALDI-TOF MS is an essential instrument in contemporary clinical laboratories, as it enables the rapid diagnosis of infectious diseases because of its cost effectiveness and high throughput [59]. The accuracy and reliability of this technology have been further enhanced by recent developments in sample preparation and database enrichment, which have resolved the challenges associated with the identification of closely related species [57,60]. In addition, MALDI-TOF MS has been shown to be capable for detecting antimicrobial resistance markers, which is essential for the management of drug-resistant infections [58,60]. MALDI-TOF MS advances with ongoing research and technological advancements, despite limitations, such as the necessity of exhaustive and up-to-date spectral databases, indicating that it will have an even greater impact on clinical microbiology in the future [59,60].

The utilization of MALDI-TOF MS in clinical microbiology is not limited to the identification of pathogens; it also encompasses the identification of antibiotic susceptibility and resistance biomarkers, thereby improving the management of infectious diseases [58,60]. MALDI-TOF MS has become the preferred method over traditional biochemical identification techniques because of its high diagnostic accuracy and rapid turnover time, which have reduced the time required to complete a microbiological diagnosis by up to 24 h [58,59]. This decrease in time is especially important for patients who are immunocompromised or have life-threatening infections [60]. By combining molecular techniques with MALDI-TOF MS, its capabilities have been further enhanced, enabling the rapid identification of microorganisms that cause hospital infections and the detection of virulence markers [57,60]. It is anticipated that MALDI-TOF MS technology will continue to play a more-significant role in the advancement of clinical microbiology, enhancing diagnostic workflows and patient outcomes through its high throughput, sensitivity, accuracy, and speed as it continues to evolve [56,57] (Figure 2D).

## 4. Enhancing Accessibility and Integration of Mass Spectrometry in Clinical Laboratories

### 4.1. Simplified User Interfaces

Efforts to develop more user-friendly MS systems that require less specialized knowledge to operate are increasingly being prioritized in the field of user interface (UI) design. Recent studies have emphasized the importance for creating digital solutions that are accessible to all users, regardless of their technical expertise, by adhering to well-established usability principles and design recommendations [61]. The process often involves user-centered design, which places the user at the core of the design process, ensuring that the interface meets their needs and preferences through methods such as user interviews, surveys, and usability testing. Simplifying the interface by focusing on essential elements and eliminating unnecessary clutter has been shown to significantly improve user experience and engagement [61]. Additionally, the use of a clear visual hierarchy, a consistent layout, and an intuitive navigation is crucial in making interfaces more understandable and easier to use. Recent research has also highlighted the need for a critical analysis of existing design recommendations to standardize best practices and enhance the usability of digital solutions across different technologies and user profiles. By synthesizing and validating these recommendations, designers can create more effective and user-friendly interfaces that cater to a diverse audience [61,62] (Figure 3A).

### 4.2. Improved Automation

Recent advances in mass-spectrometry-based proteomics have significantly improved automation, streamlining workflows and enhancing clinical applications [63,64]. Automated sample preparation has emerged as a crucial component for high-throughput and quantitative mass spectrometry analysis, addressing the tedious and time-consuming steps that can introduce analytical errors [63]. The development and optimization of workflows utilizing automated liquid-handling workstations have shown to improve speed and consistency in sample processing [63]. Integration of state-of-the-art, multi-instrument automated systems has enabled the execution of complex methods involved in mass spectrometry analysis [64]. These automated workflows facilitate increased throughput and reproducible quantitation of biomarker candidates, which are essential for processing large patient cohorts in clinical studies [63]. Furthermore, advances in automation have extended to data acquisition and processing, with the introduction of automated mass spectrometry data acquisition systems and sophisticated data analysis tools. The implementation of fully automated FAIMS-DIA (high-field asymmetric-waveform ion mobility spectrometry–data-independent acquisition) mass-spectrometry-based proteomic pipelines has demonstrated deep coverage of cellular proteomes with high throughput and reproducibility [65]. These developments in automation are particularly beneficial for clinical laboratories, enabling technicians to focus on high-value tasks and improving operational efficiency. As this field progresses, the integration of automated sample preparation, data acquisition, and analysis is expected to further enhance the adoption of mass spectrometry in clinical settings, potentially revolutionizing diagnostic testing and biomarker discovery [2] (Figure 3B).

### 4.3. Novel Ionization Techniques

Ambient ionization refers to a technique used in mass spectrometry that allows for the direct analysis of samples in their native environments without the need for extensive sample preparation. Mass spectrometry techniques enable the direct analysis of samples in their original forms with minimal or no need for sample preparation. Desorption electrospray ionization (DESI) is a commonly employed method in mass spectrometry that utilizes a charged solvent spray to remove analytes from surfaces. DESI allows for efficient and quick analyses of medicines, metabolites, lipids, and other chemicals straight from tissue slices or other sample surfaces. Another notable ambient technology is direct analysis in real time (DART), which employs a hot gas stream to desorb and ionize analytes [66]. DART has been utilized in the fields of food safety, forensics, and pharmaceutical analysis [67]. Paper spray ionization is an ambient technique that utilizes paper as a substrate for applying samples and ionization. This method enables quick and straightforward analyses of intricate mixtures [68].

Recent advancements in ambient ionization techniques, particularly paper spray ionization (PSI), have significantly enhanced the capabilities of mass spectrometry (MS) for rapid and direct analyses of complex mixtures [69,70]. PSI is a direct, rapid, and cost-effective sampling method that generates analyte ions by applying a high voltage to a small volume of solvent sprayed onto a porous substrate [69]. This technique has demonstrated the ability to provide both qualitative and quantitative MS analyses without the need for extensive sample preparation [71]. Factors such as electric fields, solvent supply rates, and paper thicknesses have been systematically evaluated to optimize the ionization efficiency, revealing that the rim–jet mode offers the highest efficiency among different spray modes [70]. The implementation of PSI in clinical settings has shown potential for point-of-care diagnostics, enabling rapid analyses of biofluids with minimal sample handling [71]. Furthermore, recent studies have highlighted the versatility of PSI in analyzing a wide range of molecules, including illicit drugs, therapeutic drugs, metabolites, and proteins, making it a promising tool for clinical applications [67]. The simplicity and efficacy of PSI, combined with its low cost and minimal biohazard and chemical waste, position it as a valuable technique for point-of-care MS analysis [69,71].

Plasma-based approaches, such as the MasSpec Pen, provide for the quick and direct investigation of tissues inside the body for purposes including diagnosing cancer during surgery [72,73]. Paper spray ionization allows for direct analysis from paper substrates, specifically for applications such as dried-blood-spot analysis [71,74]. The combination of laser-induced acoustic desorption (LIAD) with electrospray ionization enables the examination of chemicals that are susceptible to thermal degradation [75,76] (Figure 3C).

### 4.4. Different Mass Spectrometry Advantages and Disadvantages

Mass spectrometry (MS) techniques, such as ion mobility MS (IM-MS), ambient MS, and imaging MS, have become indispensable instruments in clinical chemistry. Each technique presents unique advantages and challenges. Ion mobility MS is particularly advantageous for the analysis of complex biological samples and the identification of isomers, as it enhances separation by differentiating ions based on their shape and size [9]. Consequently, it is an ideal choice for high-throughput proteomics and metabolomics. Nevertheless, its routine clinical application may be restricted by its complexity and the necessity for specialized expertise. Paper spray ionization, a form of ambient MS, enables the rapid, direct analysis of samples with minimal preparation, thereby facilitating real-time analysis and point-of-care diagnostics in clinical settings [77]. Ambient mass spectrometry may encounter constraints in sensitivity and selectivity, particularly when confronted with intricate biological matrices, despite these advantages. Spatial molecular data within tissues are provided by imaging MS, particularly matrix-assisted laser desorption/ionization (MALDI), which aids in the identification of disease-specific markers and offers insights into disease mechanisms [78]. Nevertheless, the high costs associated with this technology, as well as the necessity for comprehensive and up-to-date spectral databases, can serve as constraints [79]. Improvements in automation, data analysis, and integration with other analytical techniques are anticipated to increase the clinical utility of these MS technologies, rendering them as more accessible and efficient for routine diagnostics [79,80] (Figure 3D).

### 4.5. Integration with Other Technologies

The integration of mass spectrometry (MS) with ion mobility spectrometry (IMS) and capillary electrophoresis (CE) has significantly advanced clinical diagnostics by enhancing the separation and structural characterization of biomolecules. IMS-MS provides additional separation dimensions based on molecular shape and size, which are particularly beneficial for distinguishing isomers and improving the analysis of complex biological samples [9,10]. This hybrid technique has shown promise in clinical settings for high-throughput and high-confidence molecular characterizations, aiding in the identification of biomarkers and improving diagnostic accuracy [9]. Similarly, CE-MS combines the high-resolution separation capabilities of capillary electrophoresis with the sensitive detection of mass spectrometry, allowing for the analysis of a wide range of analytes, from small ions to large protein complexes [81,82]. This combination reduces the sample complexity and ion suppression, leading to more-straightforward data interpretation and the enhanced detection of clinically relevant compounds [81]. The use of CE-MS in clinical diagnostics has been demonstrated in various applications, including the analysis of urinary biomarkers for disease diagnoses and prognoses [81,82]. Overall, the integration of MS with IMS and CE represents a powerful approach for improving the analytical performance and clinical utility of mass spectrometry in the diagnosis and monitoring of diseases [9,10,82].

Both LC and CE are extensively employed analytical techniques, each with its own set of advantages and disadvantages. LC, particularly high-performance liquid chromatography (HPLC), is favored for its high sensitivity, robustness, and versatility, particularly when combined with mass spectrometry (MS). This makes it suitable for a diverse spectrum of applications in pharmaceuticals, environmental analysis, and food safety [83]. Nevertheless, HPLC can be expensive to operate because of the necessity for purchasing expensive solvents and apparatus, as well as the necessity of routine maintenance [84]. Conversely, CE is commended for its rapid analysis, minimal sample consumption, and high separation efficiency, rendering it as an optimal choice for the analysis of charged or polar compounds, including nucleic acids and proteins [85,86]. CE has limitations, such as lower sensitivity in comparison to that of HPLC, particularly with UV detection, and challenges in method development because of less-standardized protocols, despite these advantages [87,88,89]. Furthermore, the reduced sample-loading capacity of CE renders it as less suitable for preparative applications [86]. The decision between LC and CE is frequently contingent upon the specific analytical requirements, including the nature of the analytes, the requisite sensitivity, and the cost constraints (Figure 3E).

### 4.6. Data Analysis and Artificial Intelligence

Recent advancements in data analysis algorithms and artificial intelligence have greatly enhanced the interpretation of complex mass spectrometry (MS) data. These developments are mostly driven by recent software innovations, with new data processing tools, such as MZmine 3, MS-DIAL, and XCMS, providing improved capabilities for exploring and processing raw MS data [90]. In conjunction with these tools, machine-learning (ML) techniques offer novel approaches for interpreting clinical data, making it essential for improving biomarker discovery, disease classification, and treatment response prediction [91,92]. For instance, ML methods have demonstrated significant potential in identifying differentially expressed proteins in cancers, thus enhancing biomarker discovery [93]. Furthermore, AI applications have shown promise in clinical settings, particularly by classifying patient samples with high accuracy based on MS data, ultimately aiding in more effective disease diagnoses [94,95]. In drug response prediction, sophisticated ML algorithms are increasingly used to tailor therapies to individual patients’ profiles, marking a shift toward personalized medicine [96]. As these analytical frameworks continue to evolve, they hold the potential to redefine diagnostic processes and therapeutic strategies within clinical diagnostics [97,98,99] (Figure 3F).

## 5. Challenges and Opportunities in Implementing Mass Spectrometry in Clinical Laboratories

### 5.1. Complexity and Expertise

Despite its potential, the implementation of mass spectrometry (MS) in routine clinical laboratories faces several challenges, particularly in terms of complexity and the need for specialized expertise [100]. MS techniques require a high level of technical expertise for operation and data interpretation, which may limit their adoption in standard clinical laboratories. The intricate nature of MS, involving sophisticated instrumentation and complex data analysis, necessitates highly trained personnel, which can be a significant barrier for many laboratories [101]. Additionally, the integration of MS into clinical workflows is complicated by the need for rigorous method validation and adherence to regulatory standards, which can be resource intensive and time consuming [102]. The high costs associated with MS equipment and maintenance further exacerbate these challenges, making it difficult for smaller laboratories to adopt this technology [103]. Despite advancements in automation and user-friendly interfaces, MS systems still require significant manual intervention and expertise to ensure accurate and reliable results [5]. Moreover, the variability in sample preparation and the potential for matrix effects necessitate meticulous optimization and standardization, adding another layer of complexity to the implementation process [102]. The development of comprehensive spectral databases and robust quality-control measures is essential to overcome these hurdles and facilitate the broader adoption of MS in clinical settings [100]. Ongoing efforts to streamline MS workflows and enhance automation hold promise for reducing the technical barriers and making this powerful analytical tool more accessible to clinical laboratories [103].

### 5.2. Standardization

The implementation of mass spectrometry (MS) in routine clinical laboratories faces several challenges, particularly in terms of standardization and quality-control measures needed to ensure reproducibility and comparability of results across different laboratories [104]. The lack of standardized protocols for sample preparation, data acquisition, and analysis can result in significant variability in MS results between laboratories, undermining the reliability of diagnostic tests. Comprehensive guidelines, such as those provided by the Clinical and Laboratory Standards Institute (CLSI), are essential to establish uniform procedures and quality assurance practices that can be adopted universally [105]. The development of standardized reference materials and calibration standards is crucial for achieving consistency in MS measurements, as these materials provide benchmarks against which laboratories can validate their methods and instruments (Annotation 4). Quality control (QC) in MS-based proteomics has seen significant advances, yet the rate of adoption of these QC measures remains inconsistent, highlighting the need for more robust and accessible QC tools and protocols [106]. The establishment of metrological traceability, ensuring that measurement procedures are linked to internationally recognized standards, is another critical aspect of standardization that enhances the comparability of results across different settings [104]. Efforts by organizations, such as the Joint Committee for Traceability in Laboratory Medicine (JCTLM), to promote the global standardization of clinical test results are pivotal in addressing these challenges. Despite these efforts, the high level of technical expertise required for MS operation and data interpretation continues to be a barrier, emphasizing the need for ongoing training and education of laboratory personnel [107]. As MS technology continues to advance, the integration of automated systems and user-friendly interfaces holds promise for reducing the complexity of MS workflows and facilitating its broader adoption in clinical laboratories [3].

### 5.3. Automation

The development of automated MS-based analyzer systems is crucial for shifting from specialized laboratories to standard clinical laboratories [3]. The transition to automated MS systems offers numerous benefits, including improved efficiency, reduced human error, and the ability to bring previously outsourced testing in house [108]. Automated liquid-handling systems and data analysis solutions have been introduced by various manufacturers to streamline pre- and post-analytical processes [109]. These advancements have led to significant reductions in manual manipulations and processing times, enhancing the overall laboratory workflow [5]. The Cascadian SM clinical analyzer, a now-discontinued instrument, represents a substantial advancement in the development of entirely automated, ‘black box’ LC-MS/MS systems that can be operated without the assistance of specialized technical personnel [110]. These systems aim to integrate sample preparation, analysis, and result generation into a single automated workflow, making MS more accessible to standard clinical laboratories. However, challenges remain, including the need for a wider range of available tests and regulatory approvals. As automation in clinical MS continues to advance, it is expected to drive wider adoption of this technology in routine laboratory settings, potentially improving patient care through faster turnaround times and more comprehensive diagnostic capabilities [3,97]. Future developments in instrumentation, software, and machine-learning algorithms are likely to further enhance the clinical applications of MS and facilitate its integration into standard laboratory practices [5].

### 5.4. Cost

Mass spectrometry (MS) is a powerful analytical technique that provides both qualitative and quantitative information based on the mass-to-charge ratios of analytes. However, the cost for acquiring such sophisticated equipment is significant [111]. This high acquisition cost hinders the potential of MS analysis for a broader audience, particularly smaller laboratories that may not have the financial resources to invest in such expensive equipment [112]. Driven by the goal for expanding the accessibility and applicability of MS analysis, efforts to miniaturize MS instrumentation have been ongoing since the 1970s [111]. The past two decades have seen rapid advancements in the development of portable mass spectrometers, facilitated by improvements in microfabrication techniques, precise machining, integrated circuits, and computational modeling tools. Despite these advancements, the adoption of portable MS is still at the earliest stages, and further expansion is needed to fully realize its potential. Beyond the initial purchase, operating costs, including maintenance, calibration, and consumables, can also be significant and must be factored into the overall cost of the ownership [112]. Smaller laboratories can mitigate some of these costs by forming collaborations or utilizing shared facilities that provide access to MS instrumentation on a fee-for-service basis. Recent literature has highlighted the cost considerations in proteomic workflows, emphasizing the importance of the efficient use of instrument time and resources to ensure cost effectiveness (Figure 4).

## 6. Conclusions

Mass spectrometry has demonstrated immense potential in advancing clinical chemistry, particularly in the realms of biomarker discovery and personalized medicine. Its ability to provide accurate, sensitive, and comprehensive molecular profiles makes it an invaluable tool for modern healthcare. As technological advancements continue to address current limitations, mass spectrometry is poised to play an increasingly central role in clinical laboratories, ultimately contributing to improved patient care through more precise diagnostics and tailored therapeutic strategies. The integration of mass spectrometry into routine clinical practice represents a paradigm shift in laboratory medicine, offering new possibilities for understanding disease mechanisms, identifying novel biomarkers, and personalizing treatment approaches. As the field continues to evolve, ongoing research and development efforts will be crucial in overcoming existing challenges and fully realizing the potential of mass spectrometry in clinical chemistry.

## Figures and Tables

**Figure 1 ijms-25-09880-f001:**
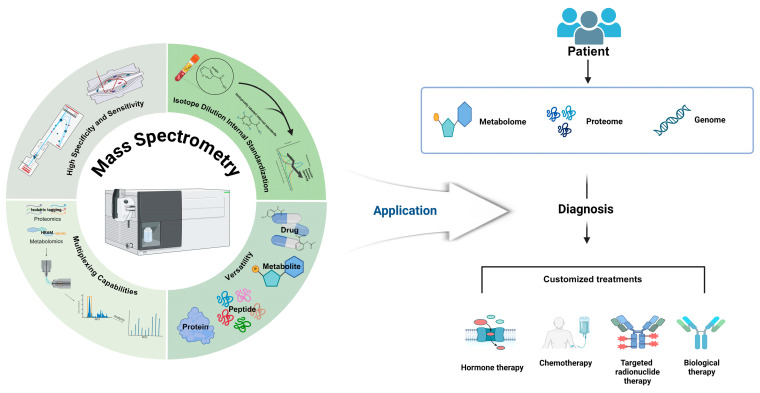
Strengths of mass spectrometry in diagnostics and personalized medicine. Mass spectrometry (MS) has significantly improved in sensitivity and specificity because of advancements in high-resolution accurate-mass (HRAM) analyzers, such as time-of-flight MS (TOF MS) and Orbitrap. These developments facilitate the accurate detection and quantification of low-abundance peptides. The integration of mass spectrometry with liquid chromatography (LC-MS) enhances separation efficiency and broadens its applicability, enabling the precise analysis of a wide range of biomolecules. This approach not only facilitates detailed proteomic and metabolomic profiling using multiplexing technologies but also allows for drug analysis, demonstrating its versatility in molecular profiling. Isotope dilution internal standardization (IDMS) further contributes to the accuracy and reproducibility of measurements by mitigating matrix effects. MS-based techniques are essential in advancing clinical diagnostics and personalized medicine, offering detailed molecular profiling that supports the understanding of disease mechanisms and the development of customized treatment strategies. This approach enables the precise and reliable analysis of biomolecules across diverse concentrations in complex biological samples, ultimately enhancing the capability for personalized medicine.

**Figure 2 ijms-25-09880-f002:**
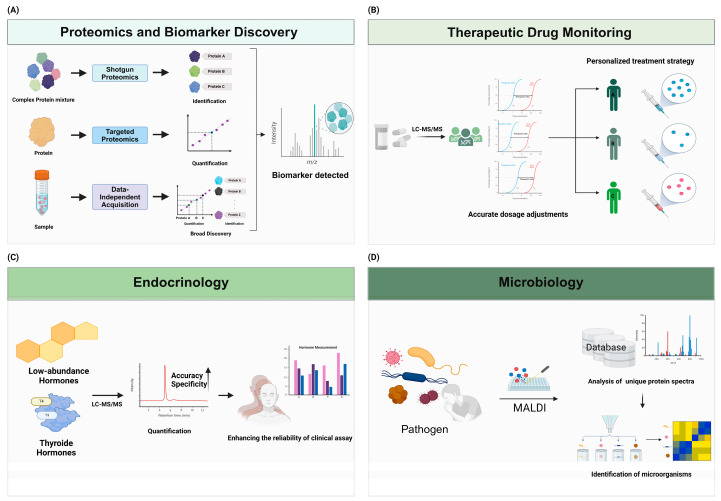
Advancements in personalized medicine and biomarker discovery through mass spectrometry. Mass spectrometry (MS) is a powerful tool for the discovery and validation of biomarkers in various diseases. (**A**) Techniques such as shotgun proteomics, targeted proteomics, and data-independent acquisition (DIA) enable the comprehensive analysis of complex protein mixtures and the precise quantification of proteins, facilitating biomarker identification and clinical applications. (**B**) The integration of MS with liquid chromatography (LC-MS/MS) enhances the accurate detection and quantification of drugs. These strengths of MS improve the efficiency and accuracy of therapeutic drug monitoring (TDM) and enable personalized treatment strategies and accurate drug concentrations. (**C**) LC-MS/MS improves the precise detection and quantification of analytes in diverse and complex matrices, including low-abundance molecules. This capability extends to the simultaneous quantification of low-abundance hormones. These technologies enhance the reliability of clinical assays. (**D**) The integration of databases with mass spectrometry (MS) technology enhances its effectiveness. By comparing unique protein spectra obtained from MALDI-TOF MS with those in comprehensive microbial databases, it enables precise pathogen identification and aids in prompt decision-making for treatments.

**Figure 3 ijms-25-09880-f003:**
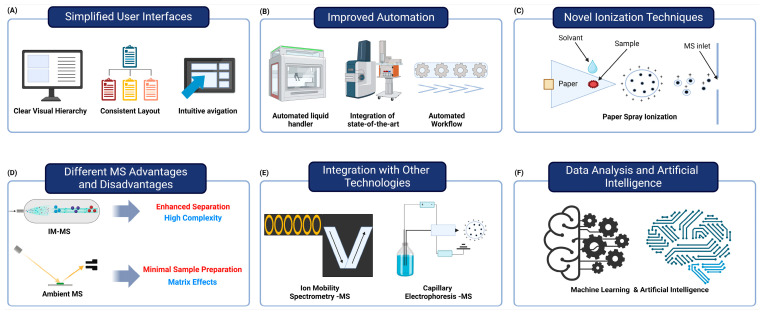
Factors contributing to the improved accessibility and integration of mass spectrometry in clinical laboratories. (**A**) The development of user-friendly mass spectrometry (MS) systems features simplified interfaces that require less specialized knowledge, improving usability. (**B**) The integration of multi-instrument automated systems and automated workflows enhances speed and consistency while reducing manual errors. (**C**) Novel ionization techniques, like paper spray ionization, enable rapid, low-cost analysis with minimal preparation. (**D**) Ion mobility MS is well suited for analyzing complex biological samples, but it is complex and requires specialized knowledge. On the other hand, ambient MS allows for simple sample preparation and rapid analysis, but it may have limitations in analysis because of matrix effects. (**E**) Integration of MS with ion mobility spectrometry and capillary electrophoresis enhances biomolecule separation and characterization. (**F**) Advanced data analysis tools and AI improve the interpretation of MS data, aiding in biomarker discovery and disease classification. These advancements make MS more accessible and efficient for routine clinical use, potentially revolutionizing diagnostic testing.

**Figure 4 ijms-25-09880-f004:**
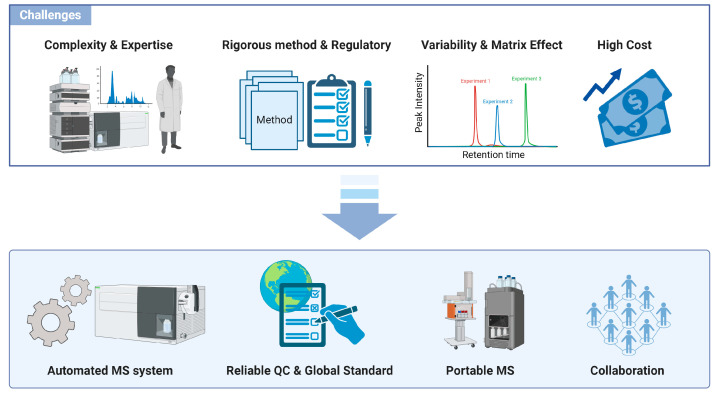
Challenges and opportunities in implementing mass spectrometry in clinical laboratories. The high level of technical expertise required for MS operation and data interpretation, combined with the need for rigorous method validation and the high cost of equipment, limits its adoption in standard laboratories. The lack of standardized protocols and reference materials creates variability in MS results, necessitating uniform procedures and quality-control measures for reliable diagnostics. Automated MS systems are making significant progress in enhancing productivity and minimizing human error, as they continue to evolve toward completely automated workflows. High acquisition and operating costs of MS equipment restrict access for smaller labs, though miniaturization and shared facilities offer potential cost mitigation strategies.

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
