# Peer review of "Mass Spectrometry Advancements and Applications for Biomarker Discovery, Diagnostic Innovations, and Personalized Medicine"

_ijms, 2024, doi:10.3390/ijms25189880_

Round 1
Reviewer 1 Report
Comments and Suggestions for Authors
This manuscript reviews the application of LC-MS in identifying disease biomarkers for diagnostic and prognostic purposes. Despite the fact that the authors have done extensive reference analyses, I think there are still significant shortcomings.
1. The title of the manuscript is ‘Mass Spectrometry in Clinical Chemistry’, however, the article is summarized in terms of LC-MS techniques. Although LC-MS is an important mass spectrometry technique, it is by no means representative of all Mass Spectrometry techniques.
2. Authors should increase the use of open ambient mass spectrometry methods and mass spectrometry imaging in clinical chemistry, which is currently an important direction in mass spectrometry technology.
3. For mass spectrometry techniques, it would be better for the authors to be able to summarize the advantages and shortcomings. of the different mass spectrometry techniques in the form of pictures, and the characteristics of their application in clinical chemistry.
Author Response
[1] Comments for the Author
The title of the manuscript is ‘Mass Spectrometry in Clinical Chemistry’, however, the article is summarized in terms of LC-MS techniques. Although LC-MS is an important mass spectrometry technique, it is by no means representative of all Mass Spectrometry techniques.
Author’s Response
Thank you for your insightful feedback. The title has been updated to be more specific, incorporating the methods employed to investigate protein dynamics: “Mass Spectrometry Advancements and Applications for Biomarker Discovery, Diagnostic Innovations, and Personalized Medicine.”
As per your recommendation, we have made significant revisions to the manuscript in its entirety. Revised and rewritten to include ambient MS, imaging MS, and overall MS, in addition to LC-MS. In our manuscript revision, we focused on the following sections: page 4, lines 154-159, page 8, lines 322-332, page 10, lines 422-434, and page 10, lines 451-456.
[2] Comments for the Author
Authors should increase the use of open ambient mass spectrometry methods and mass spectrometry imaging in clinical chemistry, which is currently an important direction in mass spectrometry technology.
Author’s Response
As per your recommendation, we have made significant revisions to the manuscript in its entirety. Revised and rewritten to include ambient MS, imaging MS, and overall MS, in addition to LC-MS. We assert the following in our manuscript revision. In our manuscript revision, on page 4, lines 154-159, page 8, lines 322-332, page 10, lines 422-434, and page 10, lines 451-456.
[3] Comments for the Author
For mass spectrometry techniques, it would be better for the authors to be able to summarize the advantages and shortcomings of the different mass spectrometry techniques in the form of pictures, and the characteristics of their application in clinical chemistry.
Author’s Response
Thank you for your feedback. We have modified figure 3 to reflect the distinctions in the application of each mass spectrometer in clinical chemistry, as well as the advantages and disadvantages, as per your recommendation. In our manuscript revision, on page 11, lines 458-476, we state:
“Mass spectrometry (MS) techniques, such as ion mobility MS (IM-MS), ambient MS, and imaging MS, have become indispensable instruments in clinical chemistry. Each technique presents unique advantages and challenges. Ion mobility MS is particularly advantageous for the analysis of complex biological samples and the identification of isomers, as it enhances separation by differentiating ions based on their shape and size. Consequently, it is an ideal choice for high-throughput proteomics and metabolomics. Nevertheless, its routine clinical application may be restricted by its complexity and the necessity for specialized expertise. Paper spray ionization, a form of ambient MS, enables the rapid, direct analysis of samples with minimal preparation, thereby facilitating real-time analysis and point-of-care diagnostics in clinical settings. Ambient mass spectrometry may encounter constraints in sensitivity and selectivity, particularly when confronted with intricate biological matrices, despite these advantages. Spatial molecular data within tissues is provided by imaging MS, particularly matrix-assisted laser desorption/ionization (MALDI), which aids in the identification of disease-specific markers and offers insights into disease mechanisms. Nevertheless, the high costs associated with the technology, as well as the necessity for comprehensive and up-to-date spectral databases, can serve as a constraint. Improvements in automation, data analysis, and integration with other analytical techniques are anticipated to increase the clinical utility of these MS technologies, rendering them more accessible and efficient for routine diagnostics.”

Reviewer 2 Report
Comments and Suggestions for Authors
This review outlines the applications of mass spectrometry to clinical chemistry. The organization of the content is acceptable dealing with the most relevant topics. Nonetheless, the authors do not take time to analyze in detail any standing development or outcome of biomarker discovery. This procedure leads the manuscript to a low academic level, rather focused on dissemination, with redundant statements and verbosity. It seems that the authors have decided to summarize what can be read in previous reviews of the issue. For instance, give some peptides which are the result of biomarker discovery using mass spec. Subsections 5.2 and 5.3 (Standarization and Automation, respectively) may serve as example of how to do.
Note: the Cascadion SM Clinical Analyzer is discontinued (https://www.360dx.com/clinical-lab-management/thermo-fisher-discontinues-cascadion-sm-clinical-analyzer).
Figures are valuable
Specific points:
1) the review is mostly focused on proteomics while mass spec is at present used extensively also in metabolomics.
2) The term "metabolites" is broad because ambiguous. Please specify which type of metabolites are referring to.
3) Subsection 4.4: give at least some pros and cons of every technology, mainly for LC and CE.
4) the last paragraph of the introduction "In conclusion, ..." is rather out of place.
5) revise the reference 3
Author Response
[1] Comments for the Author
This review outlines the applications of mass spectrometry to clinical chemistry. The organization of the content is acceptable dealing with the most relevant topics. Nonetheless, the authors do not take time to analyze in detail any standing development or outcome of biomarker discovery. This procedure leads the manuscript to a low academic level, rather focused on dissemination, with redundant statements and verbosity. It seems that the authors have decided to summarize what can be read in previous reviews of the issue. For instance, give some peptides which are the result of biomarker discovery using mass spec. Subsections 5.2 and 5.3 (Standarization and Automation, respectively) may serve as example of how to do.
Note: the Cascadion SM Clinical Analyzer is discontinued (https://www.360dx.com/clinical-lab-management/thermo-fisher-discontinues-cascadion-sm-clinical-analyzer).
Author’s Response
Thank you for your valuable feedback. As per your recommendation, we have made significant revisions to the manuscript in its entirety. We have thoroughly rewritten the report to include a comprehensive overview of the current progress and implications of biomarker discovery. In our manuscript revision, on pages 5-8, lines 205-363, we state:
“Mass spectrometry (MS) has become a potent analytical instrument in the field of clinical chemistry and biomarker discovery, allowing for the comprehensive analysis of intricate biological samples with high sensitivity and specificity. Biomarker research for a variety of diseases, such as cancer, cardiovascular disorders, and neurodegenerative conditions, has been significantly transformed by recent advancements in MS-based proteomics. In this discipline, a variety of critical MS techniques are implemented, such as shotgun proteomics for the comprehensive identification of proteins, targeted proteomics for the precise quantification of specific proteins, and data-independent acquisition (DIA) for the systematic analysis of all detectable peptides. Understanding disease mechanisms and identifying potential diagnostic and prognostic biomarkers have been significantly enhanced as a result of these methods. MS-based proteomics has been instrumental in the discovery of protein signatures that are associated with tumor progression and treatment response in cancer research. For instance, recent research has employed DIA to identify plasma protein panels for the early detection of pancreatic cancer. Large-scale plasma proteomics has revealed novel biomarkers for the prediction of incident heart failure in cardiovascular medicine. MS has facilitated the discovery of cerebrospinal fluid biomarkers that reflect pathological brain changes in neurodegenerative diseases such as Alzheimer's. Recent research has focused on the validation of blood-based biomarkers as less invasive alternatives.
MS-based metabolomics has uncovered metabolic signatures for the early detection of malignancy and the prediction of cardiovascular risk, in addition to complementing proteomics. Metabolic signatures for early cancer detection have been identified in recent metabolomics investigations, including a urine metabolite panel for bladder cancer screening. MS capabilities for clinical applications are being further enhanced by ongoing technological advancements, including ion mobility spectrometry and high-resolution mass analyzers. Standardizing protocols and validating candidate biomarkers in large, diverse patient cohorts before clinical implementation remain challenging, despite these advancements. The translation of these discoveries into routine clinical practice will be contingent upon the integration of multi-omics data and the development of simplified, automated MS workflows. The potential of incorporating metabolomic biomarkers into established risk prediction models, such as the SCORE2 model for cardiovascular risk, has been recently demonstrated, resulting in a significant improvement in predictive performance. Furthermore, the interpretation of intricate multi-omics datasets is being improved by advancements in bioinformatics and machine learning techniques, which is enabling the identification of new biomarker panels with enhanced diagnostic and prognostic efficacy. The integration of MS technologies into clinical workflows has the potential to revolutionize personalized medicine by facilitating more precise diagnosis, prognosis, and treatment monitoring across a diverse array of diseases as they continue to evolve.
3.2. Therapeutic Drug Monitoring
The gold standard for therapeutic drug monitoring (TDM) has been established by liquid chromatography-tandem mass spectrometry (LC-MS/MS) as a result of its exceptional specificity, sensitivity, and capacity to simultaneously quantify multiple analytes. This sophisticated method allows for the precise measurement of drug concentrations, which in turn facilitates the development of personalized dosage adjustments and treatment strategies in a variety of therapeutic areas, such as antiepileptics, immunosuppressants, and cytotoxic agents. The efficiency and accuracy of LC-MS/MS methodologies have been further improved by recent advancements in sample preparation techniques, chromatographic separations, and mass spectrometric detection. TDM of kinase inhibitors has garnered significant attention in the field of oncology, as a number of compounds have established exposure-response and exposure-toxicity relationships. This has the potential to enhance the efficacy of treatment and reduce the occurrence of adverse effects. The integration of TDM data with pharmacogenomic information and clinical parameters has enabled the development of more comprehensive and personalized treatment approaches, thereby improving patient care and reducing healthcare costs.
MS-based proteomics and metabolomics have emerged as potent instruments for the identification of novel disease markers and the elucidation of underlying biological mechanisms in the field of biomarker discovery. The detection and quantification of thousands of proteins and metabolites in a single analysis have been made possible by high-resolution mass spectrometry techniques, including Orbitrap and time-of-flight (TOF) analyzers, which have provided an unprecedented level of molecular profiling. Data-independent acquisition (DIA) methods have further improved the reproducibility and comprehensiveness of proteomic analyses, thereby enabling the conduct of large-scale biomarker investigations. The sensitivity and specificity of candidate biomarker validation have been enhanced by recent advancements in targeted proteomics, such as parallel reaction monitoring (PRM) and multiple reaction monitoring (MRM). MS-based methodologies have resulted in the identification of promising biomarkers for the prediction of treatment response, prognosis, and early detection in cancer research. For example, a recent study that employed DIA proteomics identified a panel of plasma proteins for the early detection of pancreatic cancer. MS has facilitated the discovery of cerebrospinal fluid and blood-based biomarkers that reflect pathological brain changes in neuro-degenerative diseases. These biomarkers have the potential to be used in the diagnostic of Alzheimer's and Parkinson's disease. The incorporation of MS technologies into clinical workflows has the potential to revolutionize personalized medicine by facilitating more precise diagnosis, prognosis, and treatment monitoring across a broad spectrum of diseases as they continue to evolve.
3.3. Endocrinology
The field of clinical endocrinology has been substantially improved by mass spectrometry (MS), which has revolutionized diagnostic and therapeutic strategies by improving the accuracy and specificity of hormone measurements. The gold standard for hormone analysis, such as steroid hormones, thyroid function tests, and vitamin D metabolite quantification, has been established by liquid chromatography-tandem mass spectrometry (LC-MS/MS) due to its superior selectivity and multiplexing capabilities in comparison to traditional immunoassays. This technology enables the precise quantification of low-abundance hormones, including dihydrotestosterone, estradiol, and aldosterone, which are frequently difficult to measure accurately with immunoassays. This has resulted in enhanced diagnostic accuracy and patient outcomes, as the high specificity of MS has revealed inaccuracies in numerous automated immunoassays, particularly under complex physiological conditions. The reliability of clinical assays has been improved by the standardization and harmonization of hormone measurements, which has been facilitated by recent advancements in MS technology. The progression of precision medicine in endocrinology has been substantially facilitated by the integration of MS-based hormone assays into routine clinical practice, which has facilitated more precise disease diagnosis and management.
MS-based proteomics and metabolomics have emerged as potent instruments for the identification of novel disease markers and the elucidation of underlying biological mechanisms in the field of biomarker discovery. The detection and quantification of thousands of proteins and metabolites in a single analysis have been made possible by high-resolution MS techniques, including Orbitrap and time-of-flight (TOF) analyzers, which have provided an unprecedented level of molecular profiling. The reproducibility and comprehensiveness of proteomic analyses have been further improved by data-independent acquisition (DIA) methods, which have facilitated the development of large-scale biomarker studies. Parallel reaction monitoring (PRM) and multiple reaction monitoring (MRM), which are recent development in targeted proteomics, have enhanced the specificity and sensitivity of candidate biomarker validation. The discovery of promising biomarkers for early detection, prognosis, and treatment response prediction has been facilitated by MS-based approaches in cancer research. For example, a recent study that employed DIA proteomics identified a panel of plasma proteins for the early detection of pancreatic cancer. MS has facilitated the discovery of cerebrospinal fluid and blood-based biomarkers that reflect pathological brain changes in neurodegenerative diseases. These biomarkers have the potential to be used in the diagnostic of Alzheimer's and Parkinson's disease. The incorporation of MS technologies into clinical workflows has the potential to revolutionize personalized medicine by facilitating more precise diagnosis, prognosis, and treatment monitoring across a broad spectrum of diseases as they continue to evolve.
3.4. Microbiology
Mass spectrometry (MS) imaging techniques offer the ability to analyze samples in a structured arrangement, providing detailed molecular information with spatial resolution. Matrix-assisted laser desorption/ionization (MALDI) imaging Mass spectrometry (MS) is frequently employed to analyze and visualize the spatial distribution of proteins, lipids, metabolites, and pharmaceuticals in tissue slices. The advancements in MALDI imaging technology have made it possible to image individual cells at a spatial resolution of less than 1 μm. DESI imaging enables the direct examination of tissues in their natural state, eliminating the need for a matrix. This technology has considerably facilitated the real-time analysis of tissues during cancer surgery, among other applications. Laser ablation electrospray ionization (LAESI) is an alternative technique that enables three-dimensional molecular imaging of plant and animal issues.
Matrix-assisted laser desorption/ionization time-of-flight mass spectrometry (MALDI-TOF MS) has revolutionized clinical microbiology by facilitating the rapid and accurate identification of microorganisms, thereby significantly reducing the time required for pathogen identification and enabling the formulation of more targeted, expedited treatment decisions. This method is capable of identifying a diverse array of microorganisms, such as bacteria, fungi, and viruses, by comparing their distinctive protein spectra to extensive databases. MALDI-TOF MS is an essential instrument in contemporary clinical laboratories, as it enables the rapid diagnosis of infectious diseases due to its cost-effectiveness and high throughput. The accuracy and reliability of this technology have been further enhanced by recent developments in sample preparation and database enrichment, which have resolved the challenges associated with the identification of closely related species. In addition, MALDI-TOF MS has been shown to be capable of detecting antimicrobial resistance markers, which is essential for the management of drug-resistant infections. MALDI-TOF MS advances with ongoing research and technological advancements, despite limitations such as the necessity of exhaustive and up-to-date spectral databases, indicating that it will have an even greater impact on clinical microbiology in the future.
The utilization of MALDI-TOF MS in clinical microbiology is not limited to the identification of pathogens; it also encompasses the identification of antibiotic susceptibility and resistance biomarkers, thereby improving the management of infectious diseases. MALDI-TOF MS has become the preferred method over traditional biochemical identification techniques due to its high diagnostic accuracy and rapid turnover time, which has reduced the time required to complete a microbiological diagnosis by up to 24 hours. This decrease in time is especially important for patients who are immunocompromised or have life-threatening infections. By combining molecular techniques with MALDI-TOF MS, its capabilities have been further enhanced, enabling the rapid identification of microorganisms that cause hospital infections and the detection of virulence markers. It is anticipated that MALDI-TOF MS technology will continue to play a more significant role in the advancement of clinical microbiology, enhancing diagnostic workflows and patient outcomes through its high throughput, sensitivity, accuracy, and speed as it continues to evolve.”
The Cascadion SM Clinical Analyzer content has been eliminated from the revised manuscript.
[2] Comments for the Author
the review is mostly focused on proteomics while mass spec is at present used extensively also in metabolomics.
Author’s Response
Thank you for your feedback. In the revised paper, we have also incorporated metabolomics as a mass spectrometer. In our manuscript revision, on page 6, lines 224-243 & pages 7-8, lines 283-320 & page 10, lines 422-434.
[3] Comments for the Author
The term "metabolites" is broad because ambiguous. Please specify which type of metabolites are referring to.
Author’s Response
Thank you for your feedback. To ensure that the metabolite type was clearly defined, we implemented modifications to the revised paper. In our manuscript revision, the urine metabolite is located on page 6, line 227, and the vitamin D metabolite is located on page 7, line 286-287.
[4] Comments for the Author
Subsection 4.4: give at least some pros and cons of every technology, mainly for LC and CE.
Author’s Response
Thank you for your valuable feedback. We have included in the updated work the advantages and drawbacks of capillary electrophoresis (CE) and liquid chromatography (LC). In our manuscript revision, on pages 11-12, lines 496-511, we state:
“Both LC and CE are extensively employed analytical techniques, each with its own set of advantages and disadvantages. LC, particularly high-performance liquid chromatography (HPLC), is favored for its high sensitivity, robustness, and versatility, particularly when combined with mass spectrometry (MS). This makes it suitable for a diverse spectrum of applications in pharmaceuticals, environmental analysis, and food safety. Nevertheless, HPLC can be expensive to operate due to the necessity of purchasing expensive solvents and apparatus, as well as the necessity of routine maintenance. Conversely, CE is commended for its rapid analysis, minimal sample consumption, and high separation efficiency, rendering it an optimal choice for the analysis of charged or polar compounds, including nucleic acids and proteins. CE has limitations, such as lower sensitivity in comparison to HPLC, particularly with UV detection, and challenges in method development due to less standardized protocols, despite these advantages. Furthermore, the reduced sample loading capacity of CE renders it less suitable for preparative applications. The decision between LC and CE is frequently contingent upon the specific analytical requirements, including the nature of the analytes, the requisite sensitivity, and the cost constraints.”
[5] Comments for the Author
the last paragraph of the introduction "In conclusion, ..." is rather out of place.
Author’s Response
In accordance with the recommendation, we eliminated the last paragraph of the introduction section in the revised paper.
[6] Comments for the Author
revise the reference 3
Author’s Response
We apologize for any errors. The reference in #3 has been rectified.

Round 2
Reviewer 1 Report
Comments and Suggestions for Authors
Accept in present form.